# B Cells and Antibodies in Kawasaki Disease

**DOI:** 10.3390/ijms20081834

**Published:** 2019-04-13

**Authors:** Michael E. Lindquist, Mark D. Hicar

**Affiliations:** 1United States Army Medical Research Institute of Infectious Diseases, Fort Detrick, Frederick, MD 21702, USA; lindquistme@gmail.com; 2Department of Pediatrics, Jacobs School of Medicine and Biomedical Sciences, University at Buffalo, Buffalo, NY 14222, USA

**Keywords:** aggresomes, antibodies, B cells, plasmablasts, inclusions, virus-like particles, endothelial

## Abstract

The etiology of Kawasaki disease (KD), the leading cause of acquired heart disease in children, is currently unknown. Epidemiology supports a relationship of KD to an infectious disease. Several pathological mechanisms are being considered, including a superantigen response, direct invasion by an infectious etiology or an autoimmune phenomenon. Treating affected patients with intravenous immunoglobulin is effective at reducing the rates of coronary aneurysms. However, the role of B cells and antibodies in KD pathogenesis remains unclear. Murine models are not clear on the role for B cells and antibodies in pathogenesis. Studies on rare aneurysm specimens reveal plasma cell infiltrates. Antibodies generated from these aneurysmal plasma cell infiltrates showed cross-reaction to intracellular inclusions in the bronchial epithelium of a number of pathologic specimens from children with KD. These antibodies have not defined an etiology. Notably, a number of autoantibody responses have been reported in children with KD. Recent studies show acute B cell responses are similar in children with KD compared to children with infections, lending further support of an infectious disease cause of KD. Here, we will review and discuss the inconsistencies in the literature in relation to B cell responses, specific antibodies, and a potential role for humoral immunity in KD pathogenesis or diagnosis.

## 1. Introduction

### 1.1. Overview

Kawasaki disease (KD), also known as Kawasaki syndrome, is the leading cause of acquired cardiac disease in children [1]. Diagnosis is purely clinical, as there are no adequately specific or sensitive tests available. The ‘classic’ diagnosis involves five days of fever and having four of the five following criteria: Mucous membrane inflammation, rash, hands and feet swelling, conjunctivitis, and a solitary inflamed lymph node mass [2,3,4,5]. If left untreated, roughly one-quarter of the children meeting clinical criteria will go on to have coronary artery inflammation, including aneurysms. Incomplete cases, those which do not fulfill four of five of the classic criteria, have similar risk of coronary aneurysms [6]. Treating affected patients with intravenous immunoglobulin (IVIG) reduces the rates of coronary aneurysms, with a minority seemingly resistant to treatment [2,3,4,7,8,9,10]. Although most aneurysms resolve, some defects are retained. Initial studies done on adults with a history of KD implies there is a greater lifetime risk of cardiac issues and early mortality [11,12,13,14]. To add to the diagnostic confusion, several infectious etiologies have also been independently associated with aneurysms [15]. It remains a frustrating diagnosis because of the unknown etiology, clinical variability, lack of specific testing, and unclear pathogenesis.

### 1.2. Genetic Background

There appears to be a genetic influence in exhibiting KD. Incidence is higher in some genetic backgrounds and consistently appears in males greater than females within those backgrounds [16]. By age five in the United States, 1 in 1000 African-American children and 1 in 2000 Caucasian children will have been affected [17,18,19]. In general, Asians have a much higher rate of KD, this is especially evident in Japanese children, whose lifetime incidence rate is near 1% [20]. This predisposition holds even for those persons of Japanese heritage raised in foreign lands, such as the United States [16].

### 1.3. Epidemiology

The etiology of KD is unknown [4,21,22]. However, there is a proposed relation to an infectious agent. Epidemiological evidence for this comes from the fact that there are seasonal peaks of KD during winter and spring months and outbreaks have been described [22,23,24,25,26,27,28]. Siblings have a higher rate of KD than the general population; usually cases are within the first year [29], and can be as high as up to 50% of cases within 10 days of each other [30]. Recent studies show a lower incidence in breastfed infants [31] and KD is rare in both newborns and individuals over five years of age. This implies a maternally derived protective immunity to a ubiquitous infectious agent [32]. This phenomenon is similar to epidemiological findings with human herpesvirus-6 (HHV-6) infections. In fact, HHV-6 is one of several potential etiologies that have been proposed as the cause of KD [11,16]. Other notable infectious agents include other Herpesviridae (Epstein Barr virus, Cytomegalovirus), human coronavirus (New Haven), retroviruses, Parvovirus B19, bocavirus, and bacterial infections such as staphylococci, streptococci, Bartonella, and Yersinia infections. Some of these agents have been independently associated with aneurysm formation [15]. Epstein Barr virus particularly is associated with coronary aneurysms [33]. Several non-infectious agents have also been proposed such as carpet shampoos, mercury and living near bodies of water [11,16]. Additionally, the recent report of tropospheric wind patterns correlating with outbreaks in Japan would not be consistent with many of the viruses that have been proposed [21,34,35]. These reports imply a relationship to an environmental antigen, as either a priming or inciting event.

If a ubiquitous childhood virus is the cause of KD, the mode of entry would likely be a common mode of infection such as fecal–oral or respiratory spread. To note, mild upper respiratory symptoms have been described in up to 35% of cases [36] with rare but more significant pulmonary disease also being reported [37]. Additionally, outbreaks in the United States have been associated with preceding viral illness [38]. Notably, however, concomitant respiratory viruses are only shown in 9% of cases [39], and in the same study that showed 35% with respiratory symptoms, 61% were noted to have gastrointestinal complaints.

### 1.4. Theories on Pathogenesis

It is possible that there is not one cause of KD, but multiple etiologies that result in similar pathogenesis. This may explain the clinical variability and lack of a definitive agent, however, the low recurrence rate even in high prevalence areas speaks against a large number of causes [40]. A superantigen response was considered by numerous groups [41,42,43,44,45,46]. Certain bacterial infections contain proteins that non-specifically bind effector cell receptors causing a more generalized polyclonal expansion and inflammation, termed a superantigen effect. Polyclonality of T cell receptor usage has been shown in KD [47,48]; however, the reports are variable as to which subset of T cell receptors are expanded [49]. Other studies support a traditional oligoclonal response consistent with an immune response against a specific etiologic agent. Oligoclonal expansion of CD8+ T cells [50] and peripheral IgM+ B cell responses have been demonstrated [5,51], and IGG+ clonality is seen in studies from our own laboratory (unpublished). Numerous other studies have not shown superantigen-associated expansions of cell subsets [50,52,53]. This concept is reviewed extensively elsewhere [45].

In a recent network and pathway analysis, responses were consistent with global activation of the immune response [54]. Although genome wide searches and similar techniques have not been definitive, genes involved with B cell activation, such as CD40 and the B lymphocyte kinase (BLK), have been identified [55,56]. There is a growing body of literature implicating specific B cell responses in the pathogenesis [57]. In this review we will focus on the literature surrounding these recent reports of antibody reactive cellular inclusions and B cell involvement.

## 2. Consideration of Humoral Immunity

### 2.1. Treatment

A number of pharmacologic agents have been used during the inflammatory phase of KD. Treatment with IVIG in KD patients can inhibit coronary aneurysm formation, implying a role for antibodies in disease pathogenesis. However, it is unclear how IVIG actually functions in this setting and if specific antibody responses are responsible for pathogenesis. Potential functions of IVIG include: Replacement for deficient specific protective antibody, anti-idiotype response against pathologic antibodies, B cell downregulation, upregulation of regulatory T cells, downregulation of neutrophil function, downregulation of dendritic cell function, and superantigen neutralization. Recent reviews have explored these functions [58,59]. The main treatment modalities used for refractory treatment are steroids, calcineurin inhibitors, and anti-TNF monoclonal antibodies; all of which have broad immunological effects [60,61]. Success with anti-TNF monoclonal antibodies seemingly argues against a significant role of B cells, as this would effectively release a suppressive action of TNF on B cell proliferation. However, calcineurin inhibition would have the opposite effect by limiting T-cell help to B cells [62]. Limited reports of treatment with anti-B cell monoclonal antibodies (anti-CD20) also support a role for B cell activation in KD pathogenesis [63]. Interluekin-1 (IL 1), has long been known to affect B cell activity [64], but it has a very broad array of inflammatory responses [58]. Notably, there is support in the lactobacillus casei mouse model for IL-1 playing a role [65]. Applicable clinical trials are listed in Table 1.

### 2.2. KD Murine Models

The first KD model system developed depended on intraperitoneal Candida injections in susceptible mouse strains [72]. This is shown to have a superantigen response mechanism. Similarly, mice developed coronary artery inflammation after intraperitoneal injection with *Lactobacillus casei* cell wall extract. Pathogenesis in this model parallels KD in that younger mice are more predisposed to develop arteritis and there is a favorable response to IVIG treatment. This disease exhibits mostly a T-cell infiltrate in coronary arterial specimens [42]. In fact, in both RAG-1 [73] and TCR-α [74] deficient mice, this arteritis is diminished [75]. Other models depend on immune complex deposition. This was observed after bovine serum albumin injection into rabbits, which exhibited a disease similar to serum sickness [76]. A number of the model systems have granulomatous changes, which have variably been seen in human specimens [77]. Presently, there is not a model system consistent with direct infectious coronary artery invasion nor that exactly replicates the pathologic changes seen in humans [78]. Considering that the cause in humans is unknown, it is unclear if any of these models of arteritis are truly applicable. Although most data from model systems are supportive of superantigen involvement, studies from human peripheral lymphocyte responses are variable and inconsistent [79].

### 2.3. Human Pathologic Studies

The lack of robust studies on human pathological studies on cellular infiltrates in KD is likely explained by the necessary reliance on autopsy specimens. A number of studies have noted lymphocytic infiltrates in samples from later timepoints. Limited studies have shown that acute infiltrates develop over time with late fibrosis occurring in the intima and adventia layers. Neutrophils seem to be the predominant initial cell infiltrate [80]. However, in a series of six specimens, neutrophil infiltration was quickly followed by lymphocyte infiltrates, then mixed lymphocyte and plasma cell infiltrates were demonstrated later, near day 19 of illness [81]. In a separate series (8 specimens), early B lymphocyte infiltration after initial neutrophil infiltrate was confirmed [80]. Prominent nodular infiltrates, similar to atherosclerotic plaque formation, have also been described, but these appear to occur at later timepoints (>3 weeks). These infiltrates consisted of T cells, macrophages, B cells and prevalent IgM+ plasma cells, with less frequent IgA+ plasma cells. The authors compare these to similar B cell rich lesions driven by both superantigens and specific infectious antigens [82].

A pathologic study on seven samples from later timepoints (most greater than two weeks after beginning symptoms of KD) revealed fewer IgM+ plasma cells compared to more prevalent IgA+ plasma cells. These were seemingly specific and prominent in seven KD biopsy specimens, however the fourteen control specimens were from autopsies that succumbed generally from non-inflammatory and non-cardiac syndromes. Notably, mature memory and immature B cells (CD20+ cells) were lacking [83]. Due to the late time point of these specimens, this may not be inconsistent with other reports reviewed previously. The lack of CD20+ B cells was theorized to be from early coronary infiltration of CD20+ B cells followed by immediate switching of these B cells to plasma cells [84]. This increase in infiltrative IgA+ plasma cells could not be explained by a generalized increase in peripheral IgA+ cell, as none was shown in acute or convalescent KD [3]. The largest study, relying on electron microscopic studies, suggests that there is an early necrotizing arteritis indicative of an acute viral infection, followed by a vasculitis, then luminal myofibroblast proliferation [77]. Although this study had 32 samples, it only had three within two weeks of disease onset and a number of findings were different than previous studies. Collection and study of these types of rare samples should continue.

Although plasma cell infiltration as outlined above is intriguing, a similar pathological response is seen in a number of inflammatory conditions such as anti- N-methyl D-aspartate receptors (NMDAR) encephalitis [85], primary sclerosing cholangitis, [86] multiple sclerosis, [87] and responses to tumors [88]. Some, such as IgA nephropathy and rheumatoid pericarditis have shown plasma cell infiltration and IgA staining [89]. In KD it is proposed that these plasma cells mature in situ from initial B cell infiltration. Monoclonal B cell infiltrates have been shown in other disorders [90]. Additionally, in situ lymphoid neogenesis is described in numerous inflammatory and infectious disease systems [91,92,93,94] and some oncologic processes [95,96]. Localized inflammation and cellular damage may lead to exposure of previously hidden self-antigens setting off a localized autoimmune cascade [97]. From pure pathological studies, it is unclear if the clonality of the IgA+ plasma cell infiltrates seen in KD represents a global inflammatory response or a specific antibody driven response against an invasive pathogen.

### 2.4. Activation of Peripheral B Cells and Antibodies

Unfortunately, there have also been few published reports on the peripheral blood dynamics during KD. Reports do show increased IgA immune complexes and levels [9], although immune complexes do not necessarily portend worse prognosis [98]. Peripheral lymphocyte analysis did not indicate an increase in IgA+ cells in acute, subacute and convalescent KD patient samples [3]. In fact, there was actually a relative paucity of IgA+ peripheral B cells from acute KD samples compared to controls. Interestingly, the lack of IgA+ peripheral B cells continued through convalescence. Other studies have shown no changes in acute and convalescent B cell subgroups, but increases in CD69+ natural killer and γδ T cells were observed [99]. Recently, the B cell marker CD 19+ was used to show an increase in both number of B cells and relative percentage in acute KD compared to controls. The percent of ‘activated’ CD86+ B cells was also significantly elevated [100]. There was also a global increase in the ability of B cells to secrete IgM, IgG, and IgA after TLR-9 stimulation, something that has been previously unexplored in the literature. Overall, in the small number of studies relating to the peripheral blood B cell compartment, there is not a consensus as to whether B cells are responsible for enhanced pathogenesis.

Although, total numbers of cells do not show consistent results, clonal expansion within the B cell compartment can be studied. A specific immune response to an agent typically has an initial inherent immune component that leads to antigen presentation to effector cells. Receptors on the effector cell surface (T-cell receptors in T cells and Immunoglobulin (IG), or antibody, in B cells) bind specific targeted areas of the agent, termed epitopes. Specific recognition by T and B lymphocytes leads to stimulation, lymphocyte replication and clonal expansion; what is termed an oligoclonal response. Oligoclonal expansion is shown in peripheral IgM+ B cells in KD [5]. Detailed pathological studies have revealed what are termed oligoclonal plasma cell infiltrates in KD arterial specimens [101], leading to the cloning of antibody J and association with the presence of the spheroid ICIs as previously discussed [102,103].

### 2.5. Cloning of Antibodies from Plasma Cell Infiltrates

Antibodies J and A were created from non-native pairing of the most prevalent sequences from reported plasma cell infiltrates (3 repeats of heavy chain and duplicates of light chains) [101,104]. On binding bronchial epithelium specimens from children with KD, antibodies J and A identified intracellular inclusions (ICI) [51,57]. In a subsequent study, 26% of the control group, composed primarily of adult patients, had similar inclusion bodies that were bound by antibody J [105]. Although many viruses can reactivate during stress (Herpesviridae family) or are considered ‘slow’ viral infections [106], the failure of numerous attempts to identify a specific infectious agent argues against such a persistent infection [57]. There remains the possibility that this is a difficult to culture virus, such as coronavirus, which had also enjoyed a short-lived consideration as the cause of KD [107].

The study that created antibodies A and J described a total of 44 heavy chain sequences and 61 light chain sequences. Other antibodies expressed, D and L, and showed no binding to ICIs. There was generally a lack of oligoclonal response with just six light chains duplicated and only the J heavy (3 times) and three other heavy chains duplicated. As these antibodies were created with non-native heavy and light chain pairings, they may have non-specific interactions [108]. This is one of the major challenges in the burgeoning bispecific antibody field [109]. Evidence of in situ maturation of antibodies, such as A and J, also does not preclude such an antibody targeting a self-antigen. Notably, two other rare clones (only one transcript each from the 44 sequence reads) showed weak binding to the same spheroid ICIs. One of these weak binding antibodies (antibody E) also bound plasma cells directly and was subsequently shown to bind kappa chains of IG. This highlights the non-specific autoimmune potential of antibodies from these types of pathologic infiltrates, which will be reviewed later.

### 2.6. Viral-like Inclusions Reported

Electron micrograph evaluation of autopsy samples from three individuals who had KD revealed ICIs and “virus-like particles” (VLPs) [103]. Unfortunately, these three structures all appeared to have different morphologies and variable association with the ICI [110]. RNA contained in KD sample ICIs was further analyzed by using laser-capture micro-dissection and subsequent 454 sequencing. No homologies to known viral RNA sequences were shown. Specifically, of the 411,561 nucleic acid reads done by 454 sequencing, only 1006 did not have significant GenBank homology [105]. This lack of homology to known viral sequences and paucity of unknown sequence is also not supportive of these being VLPs or viral aggregation of an unknown virus. The limitation to only autopsy specimens, lack of similar findings in other pathological reports, lack of VLP correlation to ICI, and lack of genetic specificity in the included RNA argues against these being related to the etiology of KD.

### 2.7. Common Structures Appear as Intracellular Inclusions (ICIs)

Because the ICIs observed in KD specimens are identified by recombinant antibodies synthesized from pathologic KD specimens, the authors of these studies conclude that these inclusions are of viral origin, and specifically related to the etiological agent of KD [51,105]. However, it is possible that these ICIs could be any number of host-derived functional structures that are typically observed as protein aggregates. Aggresomes, one such structure, are involved in shuttling of misfolded proteins during cellular stress [111]. Aggresomes are frequently observed as large intracellular aggregates of host proteins and are frequently surrounded by a “cage” of intermediate filament proteins. Available evidence suggests KD ICIs and aggresomes are distinct structures since the ICIs observed in the two autopsy specimens lacked a cytokeratin cage [103]. However, cytokeratin cages are not definitive of aggresomes and their presence may depend on the cell type. More frequently, vimentin is used as an intermediate filament marker for aggresomal cages. Because the expression of vimentin varies in bronchiole epithelial cells [112], other common markers of aggresomes could have been studied, such as ubiquitin, HSP70, HSP40, and proteasomal subunits [111]. Unfortunately, none of these markers were tested. In addition, there are several other large intracellular protein aggregates such as stress granules, p-bodies, prion-aggregates, aggresome-like induced structures (ALIS) and autophagosomes [113,114,115,116,117]. It is possible the ICI identified by antibodies A and J are one of these structures; perhaps the manifestation of KD is the improper regulation of one of these processes.

In addition to protein, the ICIs observed in the limited bronchial epithelial samples were also partially composed of RNA. While the RNA could be of viral origin it is important to note that many of the host-derived intracellular protein aggregates previously noted also contain host mRNA [113,114,116,117]. It is reasonable to conclude that if the ICIs observed in KD patients were related to one of these structures, they would positively stain for RNA.

### 2.8. Anti-Self-Antibody Responses

As reviewed, the similarly cloned antibody E was shown to bind a self-antigen [102]. The autoimmune aspects of KD have recently been reviewed [118]. Self-antigen responses to a variety of targets have actually been well described in KD. These include recent reports of antibody responses to type III collagen, myosin [119], cardiolipin [120], alpha-enolase [121], and anti-endothelial antibodies. Anti-endothelial antibodies are particularly interesting as these are seen in other disorders, such as SLE and renal allograft rejection [122]. Other vasculitides have also been associated with anti-endothelial antibodies. These have been shown to cause upregulation of E-selectin, VCAM-1, ICAM-1 and NFκB [123]. Responses to these antibodies include upregulation of inflammatory cytokines and apoptosis of the endothelial cells.

In KD subjects, a polyclonal antibody response against endothelial cells has been described [124]. Cytokines, such as IFN-γ, IL-1 and TNF, that would be present during generalized inflammation, facilitate a pathological anti-endothelial response of circulating IgG and IgM antibodies associated with acute KD [125,126]. In cell lysis assays, pathogenesis was eliminated by clearing the serum through anti-IgG and anti-IgM sepharose columns supportive of no role of peripheral anti-IgA responses. This does not eliminate the potential role of intra-tissue IgA+ plasma cell development in pathogenesis as has been postulated [83,102]. Other studies support significant IgM mediated cytotoxicity against endothelial cells in KD patients [127]. Prevalent IgM anti-endothelial responses in KD have also been shown without cytokine stimulation [127,128]. In a mouse model system, anti-endothelial antibody responses were replicated, but these did not demonstrate cardiac vascular involvement [129]. The case report of anti-B cell monoclonal antibody success was proposed by the authors to be due to the downregulation of such an anti-endothelial invasive effect [63]. Although intriguing, it remains unknown if these anti-endothelial responses actually contribute to the vasculitis in KD and other vasculitides [123].

### 2.9. Similar Plasmablast Responses in KD and other Infections

Recent data from our laboratory further supports an infectious disease etiology playing a role in KD. Numerous studies show that after an antigenic challenge, vaccination and natural infections, B cells transitioning to plasma cells, termed plasmablasts (PBs), can be seen in the peripheral blood [130,131]. These can be recognized by surface markers of CD19, downregulation of CD20, and high levels of CD27 and CD38 [3,132]. In comparison to the general circulating B cell population, PBs are enriched for B cells that contain infection-specific antibodies [133,134]. This is variable as some studies show massive and high enrichment of PBs targeting the antigen of interest [135,136], while other studies show polyspecificity of the PB population and limited enrichment [137,138,139,140]. Immunization studies in adults (tetanus [141], influenza [132], and rabies [142]) show PB have more consistent enrichment for specific antibodies, temporally peak 5–10 days after immunization, and are predictive of later sero-immunity [143]. Elevated circulating peripheral PBs are not specific to infections, as they are elevated in a number of autoimmune diseases and their levels correlate to disease flares [144]. Although certain infections, such as dengue virus, may set off exceedingly high PB levels [145], PB quantities tend to be significantly higher in autoimmune conditions than levels achieved during vaccination or post-infection. This excessive circulating PB response corresponds to flaring of the underlying inflammatory disease, and specifically correlates with c-reactive protein (CRP) level in studies on ulcerative colitis [145,146] and IGG4 related disease [147,148].

We postulated that if KD is caused by an infection, we should observe a predictable rise of PBs in the peripheral blood. We collected samples from 18 children with KD and 69 febrile controls presenting to the emergency department. Overall, we saw an increase in IGG+ B cells, but not a cumulative increase in B cells [149]. Notably, we did not observe an increase in circulating IgA+ B cells. The result of this study is consistent with the majority of the literature that shows B cell stimulation and increasing peripheral B cell numbers during KD [4,5,99,100]. Both KD and infectious control children showed comparable elevations of PBs compared to controls [149]. Importantly, the levels did not correlate with CRPs and were not excessive, which are characteristics of PB responses in autoimmunity. Unfortunately, only five children had repeat samples. Of these five, all had PB elevations on one or both timepoints, leaving only 3 of 18 KD samples not having a measurable elevation of their PBs in this study generally limited to one timepoint. We are currently collecting samples over multiple timepoints to more thoroughly explore this phenomenon.

Ongoing studies are exploring heavy and light chain usage in B cells and PBs during KD with next generation sequencing techniques. To specifically target an etiology, we have created monoclonal antibodies with pairing of the heavy and light chains utilizing the 10x Genomics^®^ Single Cell sequencing technology [150]. As an example, from the PB rise of near 11% of circulating B cells seen in subject 24, we created a panel of 946 paired heavy and light chain sequences. From this sample, there are a number of clones with exact sequence repeats (roughly 5%). To assign clonal groups, we analyzed the sequences using CDR3 length and sequences, and compared predicted germline antibody sequences (from IMGT, [151]). Roughly 40% of clones can be assigned to have clonal relationships. Several of the monoclonal antibodies we generated are presented in Table 2. We chose these fifteen antibodies to highlight markers that show somatic hypermutation and clonal expansion. These are highlighted by predicted clonal members, isotype switching, nucleotide substitutions from predicted germlines, and increases in the subgroup replacement to silent nucleotide mutation (R/S) ratios. Elevation of this ratio supports clonal selection of affinity matured antibodies which would correlate with an increase in mutations leading to changes in the amino acids, particularly in the antigen binding complementarity determining regions (CDRs) [152]. Work on identifying the protein targets of these antibodies is ongoing. Because of the characteristics seen in these antibodies, we hypothesize these antibodies target the etiology that caused KD in this child.

## 3. Discussion

The roll of B cells and plasma cells in KD is controversial (summarized Table 3). Much of the B cell and antibody data reviewed herein show inconsistent, contradictory or unsubstantiated findings. Pathological specimens and model systems are variably supportive or inconsistent with what is known from human studies. Although B cell and plasma cell infiltration in pathology specimens is intriguing, whether they are bystanders activated by a superantigen effect, are responding to a self-antigen revealed by inflammation, or specific against an infectious etiology, is currently unknown. Like the mouse models and attempts at developing new therapeutics, it is hard to be confident in any one approach without knowledge of the etiology. Although published B cell studies relating to KD are somewhat inconsistent, recent data using advanced sequencing techniques show promise for identification of an etiology.

Since KD is a clinical syndrome without a definitive marker of diagnosis, many of the studies may be influenced by “generous” case definitions of the study participants. Most of the studies reviewed do not include detailed clinical information or rigorous case definitions. This is a general problem with the literature in this field; these machinations seem more consistent in clinical trials and epidemiological studies but are rare in bench-science studies. This potential selection bias may be negatively influencing reproducible, definitive findings and conclusions.

This is a rich opportunity for clinical investigators. Rigorous studies are needed on those children who present with KD. If any pulmonary findings are found, bronchial washings should be obtained and stored for potential molecular diagnostics. Other samples, such as peripheral blood mononuclear cells and serum, should be taken and banked for future studies. Thorough autopsy evaluation should be pursued on any subjects who succumb during the acute or convalescent phases of KD. Improved reporting and national registries would go a long way in establishing a representative pool of patients. Studies currently ongoing on peripheral cytokine profiles, B cells and PBs may show a consistent marker to help define who has KD. A correlative diagnostic marker, possibly even antibody derived, would be a highly desirable first step in future studies.

## Figures and Tables

**Table 1 ijms-20-01834-t001:** Advanced clinical trials for treatments to prevent coronary aneurysms in Kawasaki Disease.

Drug	Clinical Trials	Phase	Status	Closure Date	Results Summary or Comments
Infliximab	NCT02298062	3	completed	September, 2015	
Infliximab	NCT00760435	3	resulted	October, 2012	Improved defervescence, well tolerated, variable z- score reduction [66].
Infliximab	NCT01596335	3	resulted	October, 2014	Improved defervescence, well tolerated [67].
Infliximab	NCT03065244 “KIDCARE”	3	recruiting	September, 2020	
Etanercept	NCT00841789	2	Active, not recruiting	August, 2018	
Anakinra	NCT02179853	2	recruiting	December, 2020	
Anakinra	NCT02390596 “Kawakinra”	2	Recruiting	April, 2019	
IVIG doses	NCT00000520	3	Completed	November, 1989	Single dose of IVIG is better than splitting doses [68].
IVIG + pulsed steroids	NCT00132080	3	Completed	March, 2005	No difference, refractory lower number than expected [69].
IVIG 1 g or 2 g	NCT02439996	3	Completed	September, 2016	
IVIG + 5 days prednisolone	NCT03200561	3	Recruiting	December, 2020	Proposal published [70].
IVIG without Aspirin	NCT02951234	na	Recruiting	August, 2019	Proposal published [71].

Na- not applicable; z- score- standard deviations from the mean.

**Table 2 ijms-20-01834-t002:** Putative anti-Kawasaki disease etiology antibodies.

Monoclonal Antibody Clones	Clonal Members	Exact Replicants	IG Isotypes	VH ^ CDR3 Length	Nucleotide Substitutions (%)	VH R/S * CDR1	VH R/S * CDR2	VL ^&^ CDR3 length	Nucleotide substitutions (%)	VL R/S * CDR1	VL R/S * CDR2
24-01	10	5	G1	19	97.8	0	1/0	11	97.7	7	0
24-02	4	4	M; G1	19	93.8	2/0	5	9	95.0	3/0	0
24-25	6	2	G1,3	17	94.4	5/0	3/0	9	95.3	2/0	2/0
24-29	6	2	G1,2,3	11	85.6	4.5	14/0	8	85.8	5	4
24-39	5	2	G1	15	90.6	2.3	2.1	11	94.6	2	8/0
24-49	3	2	G2	20	93.4	1.5	5	13	96.5	0	2/0
24-67	5		G1	18	93.5	4.3	19.5	9	97.0	9/0	5
24-377	4		G1	14	92.3	8.0	6.0	9	96.2	9.0	0.5
24-439	2		M; G2	11	91.5	4.0	2.0	11	94.4	2/0	2/
24-441/659	15		M; G1	11	93.1	31.0	30.0	9	97.6	6.5	18/0
24-595	5		M; G1,2	15	93.0	9.5	13/0	9	95.6	2.7	0
24-815	8		M; G1	15	95.4	5.5	9.0	10	97.7	4	9/0
24-893	4		G1	12	91.7	2.3	2.4	10	94.4	19/0	8
24-908	3		M; G1,3	20	96.5	4.0	2.5	9	98	5	2

^ VH- heavy chain variable region, ^&^ VL light chain variable region, * Replacement to silent nucleotide mutation ratios (R/S).

**Table 3 ijms-20-01834-t003:** Possible ways humoral immunity plays a role in KD.

	Possible Importance	Contrary Findings and Considerations
Efficacy of IVIG	Theoretically can provide antibodies to specific etiology	Function in KD theoretical, many different potential functions of IVIG
Treatment with anti-CD20 antibody	Directly downregulates IG production	Limited reports and no prospective trials
Response to IL-1 inhibitors	Downregulates IG production, mouse models support IL-1 role	Many other broad affects
Coronary plasma cell infiltrates	Seen on coronary path specimens, theorized direct response to infectious agent	Plasma cell infiltrates also seen in autoimmune disorders
Anti-self antibodies	Can cause apoptosis of endothelial cells	Later finding, not universally seen; unclear if part of etiology or response to tissue damage
Plasmablast (PB) level	Level similar to infection, may be set off by etiology of KD	Number of coinfections and IVIG may make defining specificity difficult
PB timing	Similar to that of infection	Pure autoimmune has PB rise, but often higher/flare correlated

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
