# Peer review of "B Cells and Antibodies in Kawasaki Disease"

_ijms, 2019, doi:10.3390/ijms20081834_

Round 1
Reviewer 1 Report
excellent work that summarizes the salient points of the studies so far carried out to explain the pathogenesis of KD. It therefore represents a starting point for further reflections and proposing new study strategiesAuthor Response
thank you, we hope we are at a good starting point.
Reviewer 2 Report
This work is a well researched comprehensive review about B cells and antibodies in Kawasaki disease.
I have no objection to publication.
Minor Points:
In line 173 I felt some introduction to the analysis of specific antibodies was required. This introduction came in the paragraph following line 246. Maybe this could be pulled forward.
Sentence in line 267 is hard to understand without looking in the references.
Author Response
Thank you for taking the time to review our manuscript and your comments, we have addressed them as such-
In line 173 I felt some introduction to the analysis of specific antibodies was required. This introduction came in the paragraph following line 246. Maybe this could be pulled forward.
Thank you, section 2.7 has been moved forward, now 2.4, and this portion has been renumbered.
Sentence in line 267 is hard to understand without looking in the references.
This has been clarified.
Reviewer 3 Report
In this review by Lindquist M and Hicar M, authors elegantly present arguments to consider in Kawasaki disease (KD), the important role of B cells, antibodies and infectious diseases.
The review is clear and up-to-date.
- Authors are invited to present a table to summarize arguments supporting a role of B cells and autoantibodies in KD.
- IVIG act through multiple mechanisms then limiting such activity to the restauration of the humoral immunity is limited.
- The part related to the treatment may be improved, and a figure may help the reader.
- Fiew typos
Author Response
Thank you for taking the Time to read our manuscript and comment. We have addressed you suggestions as follows:
- Authors are invited to present a table to summarize arguments supporting a role of B cells and autoantibodies in KD.
A table of the key topics covered in this review has been added.
- IVIG act through multiple mechanisms then limiting such activity to the restauration of the humoral immunity is limited.
We added a sentence to explain the numerous functions of IVIG that may be in play. This was previously just referenced as “Recent reviews have explored these functions [58, 59]”
.
- The part related to the treatment may be improved, and a figure may help the reader.
This section has been expanded. We have added a number of points from this section to the summary table (suggested above).
- Few typos
We have thoroughly reviewed the manuscript and corrected a number of minor typos. A tracked changes version and final version were submitted.